# Computation of backwater effects in surface waters of lowland catchments including control structures – An efficient and re-usable method implemented in the hydrological open source model Kalypso-NA (4.0)

Sandra Hellmers[1], Peter Fröhle[1]

[1]Institute of River and Coastal Engineering, Hamburg University of Technology, Hamburg, 21073, Germany

*Correspondence to*: Sandra Hellmers (s.hellmers@tuhh.de)

**Abstract.** Backwater effects in surface water streams as well as on adjacent lowland areas caused by mostly complex drainage and flow control structures are not directly computed with hydrological approaches, yet. A solution of this weakness in hydrological modelling is presented in this article. The developed method enables to transfer discharges into water levels and to calculate backwater volume routing along streams and adjacent lowland areas by balancing water level slopes. The implemented and evaluated method extends the application of hydrological models for rainfall-runoff simulations of backwater affected catchments with the advantages of (1) modelling complex flow control systems in tidal backwater affected lowlands, (2) less effort to parameterise river streams, (3) directly defined input factors of driving forces (climate change and urbanisation) and (4) runtime reduction of one to two orders of magnitude in comparison to coupled hydrodynamic models. The developed method is implemented in the open source rainfall-runoff model Kalypso-NA (4.0). Evaluation results show the applicability of the model for simulating rainfall-runoff regimes and backwater effects in an exemplary lowland catchment (Hamburg, Germany) with a complex flow control system and where the drainage is influenced by a tidal range of about 4 m. The proposed method is applicable to answer a wide scope of hydrological and water management questions, e.g. water balances, flood forecasts and effectiveness of flood mitigation measures. It is re-usable to other hydrological numerical models, which apply conceptual hydrological flood routing approaches (e.g. Muskingum-Cunge or Kalinin-Miljukov).

## 1    Introduction

Open demand exists in hydrological modelling of rainfall-runoff regimes in backwater affected lowlands. The flow routing in lowland catchments is characterised by artificially drained catchments using manifold flow control structures. The occurrence of backwater effects in such complex lowland river streams as well as on adjacent lowland areas pose an open research question in hydrological modelling. Adjacent lowland areas in this article are distinguished by a low ground level and connection to rivers. The size of lowlands varies from narrow riparian areas, wetlands, shallow retention spaces, floodplains or vast partly urbanised marsh- or swamplands. Hydrological models are applied to simulate processes in the compartments of the (1) surface-atmosphere interaction, (2) the transition between soil-vegetation-atmosphere, (3) the processes in the vadose zone of the soil and (4) the flood routing in the receiving surface waters. In lowlands, the last two issues require more detailed considerations because of mostly high groundwater levels and the drainage against fast changing water levels in tidal streams of complex flow control systems. For simulating the interaction between groundwater and surface water quite a few approaches are available (Brauer et al., 2014; Waseem et al., 2020; Sun et al., 2016). However, modelling backwater effects in tidal streams with fast changing water levels in complex flow control systems of lowland catchments directly with hydrological models is not implemented in most hydrological approaches up to now (Waseem et al., 2020).

Simulating backwater effects, velocity fields and the spatial distribution of water depths for flood inundation maps demands for 2D or 3D hydrodynamic-numerical models with the numerical integration of the partial differential equations describing the flood routing processes. To compute spatial detailed simulation results in river streams and flood plains, coupled

hydrological and hydrodynamic model approaches fit well to meet the required modelling objectives. But, hydrodynamic-numerical models require larger effort to parameterise river streams and simulation times, which are at least one to two orders of magnitudes longer in comparison to conceptual hydrological flood routing approaches to model river streams. High resolution data describing the topography of the main channel and the natural flood plain in the case of bank overflow is necessary. Hence, the availability of suitable detailed profile data from measurements is significant for hydrodynamic-numerical modelling. The larger effort in data resources and runtime for hydrodynamic-numerical model simulations is no limitation for answering special research questions and to create detailed inundation maps. However, applying a coupled hydrological-hydrodynamic model shows disadvantages in the application on meso to regional catchment scales (>100 km²) and for operational forecast applications. Therefore, it is proposed in this article, that a stand alone hydrological approach can be beneficial in flood forecasting models to enable parsimonious and efficient modelling of flood routing and backwater effects in lowlands, by a conceptual hydrological method producing less detailed results.

The demand to solve this weakness in hydrological numerical models increases, since in low lying tidal catchments, the pressure on current storm water flow control systems raises due to combined impacts of enlarged urbanisation on the one hand and climate change induced sea level rise in combination with heavy storm events on the other hand (IPCC, 2013b, 2013a; UN DESA, 2018). Studies about the combined risk of high tides (storms) and stormwater events are given by (Lian et al., 2013; Nehlsen, 2017; Klijn et al., 2012; Zeeberg, 2009; Huong and Pathirana, 2013; Sweet et al., 2017). These selected examples all show a conformity about the tendency that low lands will be faced by higher pressures to mitigate flooding in the future. A promising flood mitigation measure against the effects of (high) precipitation events in low lying catchments is the controlled temporary storage of water in retention areas. However, state-of-the-art hydrologic approaches reveal shortcomings in modelling the flood routing and retention volume in backwater affected lowland catchments.

### Objectives

To resolve the afore described shortcomings in hydrological approaches to model the flood routing in backwater affected lowland catchments five objectives are defined. The method shall be (1) applicable to model complex flow control systems in backwater affected lowlands, (2) efficient by using short run-times for real-time operational model application, (3) open for further model developments, (4) re-useable for other hydrological model solutions and (5) parsimonious with regard to the complexity of input parameters. Reaching a balance between model structure details (namely complexity) and data availability is an important issue to keep the model as parsimonious and efficient in runtime as possible, but complex enough to explain the heterogeneity in the areas and the dynamics in the hydrological processes. To accomplish the defined five objectives for a re-usable, open, efficient and parsimonious hydrological method to model backwater effects, the authors suggest to develop a conceptual extension approach for state-of-the-art flood routing methods (for instance Muskingum-Cunge or Kalinin-Miljukov).

### Outline

The literature review in section 2 discusses current weaknesses in hydrological models to simulate backwater effects and subsequent flooding of adjacent lowland areas. The theoretical concept in section 3 and the developed method in section 4 explain the worked out solution. The implementation of the methodology is realised in the open source hydrological model Kalypso-NA version 4.0 (section 5). The evaluation of the method is done using observed data of an exemplary lowland catchment study in Hamburg, Germany, where a complex drainage system and backwater affected streams have a significant impact on the flow regime (section 6). A discussion of results point out the main findings and limitations in section 7. The article closes in section 8 with a summary and an outlook on follow-up research.

## 2    State-of-the-art in hydrological modelling to compute flood routing and backwater effects in lowlands

Flood routing describes the processes of translation and retention of a flood wave moving along a stream in downstream
direction. To simulate the flood routing in rivers different approaches are applied: (1) pure black box (namely empirical,
lumped), (2) hydrological conceptual or (3) hydrodynamic-numerical approaches (Maniak, 2016; Hingray et al., 2014). The
applicable flood routing method needs to be chosen with respect to the modelling purpose and available data. Computation of
water depths and backwater effects in rivers as well as on forelands by using hydrological approaches (1 and 2) is rarely done
and up to now mostly linked with comparatively high uncertainties. The missing applicability of hydrological approaches for
simulating backwater effects is shown in a recent study within the North German lowlands (Waseem et al. 2020).

Commonly applied conceptual hydrological approaches are the 'storage routing' by Puls (1928), 'Muskingum' or
'Muskingum-Cunge' routing described by McCarthy in (1938) or (Cunge, 1969), 'Kalinin and Miljukov routing' (1958) or
'linear reservoir and channel cascade routing' presented by Maddaus in (1969). The purpose of hydrological flood routing
approaches is to compute the discharge hydrographs in the considered stream segments. For hydrological approaches,
conceptual or empirical parameters are calibrated based on observed events like in the frequently used Muskingum method. A
compromise are hydrological methods using profile data of streams to model the flood routing, for example in the Muskingum-
Cunge approach (Cunge, 1969) as well as the approach of Kalinin and Miljukov, 1957. These concepts use profile information
in a conceptual way and require far less calculating effort for meso scale modelling (> 100 km²) than hydrodynamic numerical
approaches.

Only few related studies are available with respect to model backwater effects in meso scale catchments with hydrological
approaches, while none of the reviewed studies enabled the computation of backwater retention in lowland areas for mitigating
backwater induced flooding. Coupled hydrological-hydrodynamic computation models like in MIKE SHE coupled with MIKE
11 (Waseem et al., 2020) or in the German Model NASIM coupled with a hydrodynamic computation model (Loch and Rothe,
2014; Dorp et al., 2017) are not part of this comparison, because of the afore described disadvantages in hydrodynamic
approaches. A focus is set on direct or stand-alone hydrological model enhancements.

In (Waseem et al., 2020), a review of models is published with regard to simulate important hydrological processes in coastal
lowlands. This review shows weaknesses in the model SWIM (soil and water integrated model) and HSPF (hydrological
simulation program—FORTRAN). The approaches in the models SWAT (soil and water assessment tool) und MIKE SHE
show good conformity to simulate processes in lowlands while both are not applicable to model backwater effects in the river,
on floodplains or other adjacent lowlands and backwater effects caused by control structures (sluices, pumping stations and
tide gates). An enhanced approach in SWAT for riparian wetlands (SWATrw) is presented in (Rahman et al., 2016) to compute
the surface water interaction between river streams and explicitly defined wetlands, while backwater effects in streams are
unconsidered. The modified SWAT-Landscape Unit (SWAT-LU) model enables to compute horizontal hydraulic interactions
between a river and the aquifer beneath the adjacent floodplain (Sun et al., 2016). Similarly, in the Rainfall-Runoff Modell
WALRUS (Wageningen Lowland Runoff Simulator) a lumped approach is realized to model the following processes: (1)
groundwater–unsaturated zone coupling, (2) groundwater–surface water feedbacks and (3) seepage and surface water supply
(Brauer et al., 2014). These are important model features to model the runoff regime in lowlands, but neither of the approaches
enable to compute backwater effects (1) along streams, (2) among stream sections and the land surface and (3) in river sections
influenced by upstream of control structures.

More national specific studies to model backwater effects in streams are done with the German model ArcEGMO (by the
'Büro für Angewandte Hydrologie', Berlin). The hydrological model 'ArcEGMO' takes into account backwater effects by
hindering the downstream flood routing when the water level at the downstream segment is higher than the upstream one
(Pfützner, 2018). The method presented by National Hydrological Forecasting Service (NHFS) in Hungary (Szilagyi and
Laurinyecz, 2014) applies a discrete linear cascade model to account for backwater effects in flood routing by adjusting a
storage coefficient of the cascade. The ArcEGMO and NHFS method calculate a retained flood rooting, but neither computes

backwater volume being routed into upstream segments by a reverse flow direction nor the backwater induced flooding of adjacent lowland areas.

In a study by (Messal, 2000), backwater effects among river streams and the subsurface flow in river banks are modelled exemplarily for the catchment Stör (1157 km²) in Schleswig-Holstein, Germany. Messal applies a proportional relationship between upstream and downstream elements for calibration purposes. The model serves well for the catchment study Stör, but the parameter values are non-transferable to other catchments because of a lack in physical descriptions.

Another approach is presented by (Riedel, 2004) to model the backwater effects among river streams in German lowlands on the example of two tidal tributaries of the Weser river. The approach uses the reservoir cascade theory including the input parameters of the roughness coefficient by Manning-Strickler and geometric descriptions of the profiles for the flood routing computation. The river is modelled as a cascade of reservoirs (namely a NASH-cascade), while the water level from the previous time step of the downstream segments are taken into account to compute the flood routing. A time step shift in the computational approach is accepted by (Riedel, 2004) because he reduced the simulation time step size to one minute. The model computes a reservoir cascade on the basis of a defined boundary condition at the downstream segment. However, the explicit simulation of backwater induced flooding of flood prone areas or adjacent lowland areas is not included.

These reviewed hydrological methods compute backwater effects in a more or less conceptual way with the described weaknesses and limitations. None of these studies analysed the backwater induced flooding of lowland areas or in this specific case, retention areas. Consequently, none of the studies accomplish to simulate a controlled retention of backwater volume in such areas, a subsequent drainage and neither the computation of hydrological processes influenced by backwater induced flooding. Further on, most studies do not apply physical-based parameters to transfer validated values and knowledge from one catchment to other studies. A methodology to solve these shortcomings is proposed in this article.

## 3 Theoretical approach to enhance a hydrologic conceptual flood routing method to compute backwater effects

To reach the described objectives, a state-of-the art conceptual hydrological method is extended to be applicable for the computation of backwater effects in streams and adjacent lowland areas (incl. retention areas). This section describes the theory of the conventional hydrological approaches to compute the flood routing (3.1), the concept of modelling control structures in tidal lowlands (3.2) and the approach to compute backwater effects with a conceptual hydrological approach in streams and adjacent lowland areas (3.3).

### 3.1 Conceptual hydrological flood routing approach

State-of-the-art hydrological flood routing theory in free flow conditions describes the flood wave propagation in streams which are not affected by downstream conditions. This means that an afflux in front of obstacles downstream of the considered stream segment is assumed to have no impact on the upstream segments. With this assumption, backwater effects are not considered. Flood routing processes depend on the characteristics of the drainage network comprising the geometry of profiles, gradients and roughness of the streams. Linear or non-linear Muskingum approaches have no physically based parameterisation and require input parameters, which are based on observed data in upstream and downstream segments of rivers. Therefore, these hydrological approaches are not suitable for the simulation with changed geometries or changed flow conditions in streams where no observed data is available. This lack is solved in two approaches, which are based on physical characteristics such as river geometry, stream length, roughness coefficient and river bed slope. On the one hand, the Muskingum–Cunge (often used in the United States) and on the other hand, the Kalinin-Miljukov (KM) flood routing approach are applicable. For this work, the approach of Kalinin-Miljukov is chosen, since this approach is widely applied in Germany and Eastern Europe.

The approach of (Kalinin and Miljukov, 1957) (KM-approach) divides a stream into a number of characteristic lengths. Each length is considered to be short enough for assuming a quasi-stationary relationship on the basis of a hysteresis curve. Different

derivations of the KM-approach are given in literature and are discussed for example by Koussis [2009]. More details about the applied approach in this work is explained in the *suppl. section 4*.

With such conceptual hydrological flood routing approaches the magnitude and time of flow along a stream on the basis of stream characteristics is determined. It describes the (free flow) propagation of discharge through streams, whereby translation and retention processes along the stream change the shape of the hydrograph from an upstream to a downstream point. The explicit direction of computation from upstream to downstream in flood routing approaches limits to include effects derived from downstream obstacles. Backwater effects caused by an afflux are ignored in these conceptual hydrological approaches and an extension is therefore developed in this article (see section 3.3).

### 3.2   Concept to model control structures in lowland catchments

Backwater effects in river sections are often caused at obstacles like weirs, (tide) gates, retention or detention reservoirs, which also function as control structures in streams. It is required to model these structures in hydrological models since such control structures are regularly used to control the flow in catchments. In this article, we focus on control structures frequently applied in lowland drainage areas. Operation rules of control structures are mostly pre-defined depending on operative criteria. The criteria are normally based on thresholds of water level, discharge or precipitation intensity within hindcasted or forecasted data (see Fig. 1). Since the data time series influence the status of control structures, they are defined in this article as drivers. There is a difference between pre-set and on-the-fly processed driver data. Pre-set data time series are imported such as observed water level or precipitation data. Additionally, data series which are computed during runtime (e.g. discharge) can serve likewise as drivers and are processed on-the-fly.

When a threshold of an operative criteria is reached during the runtime of the model, the status of the system is changed (e.g. opening or closing a gate). The change of the status based on reached thresholds is described in control functions, which are checked per time step. In a control structure the retained water can cause backwater effects in upstream direction if an afflux of water occurs. Control structures are one component type within a hydrological network. Other component types are streams (linear data structures), areas (spatial data structures) and nodes (point data structures). An explanation of these components of a hydrological network is given in the supplement (*suppl. section 3*).

### 3.3   Concept of the flood routing enhancement to compute backwater effects

The afore described hydrological conceptual approach (here, of Kalinin and Miljukov) is enhanced by using the resulting water level, volume and discharge (WVQ) relation to compute backwater effects per stream element. The concept enables to compute a backwater volume routing according to the water level slope. This is illustrated in a scheme in Fig. 2 for a river longitudinal segment which is separated in several strands. At the downstream segment a control structure is located. In stage (1) the free flood routing in downstream direction is computed. When the barrier (e.g. a tide gate) is closed by control functions (stage 2), an afflux of water is generated (stage 3). The afflux initiates a 'backwater volume routing' (stage 4), meaning that the water volume is routed in upstream direction to equalise the surplus water level of the afflux. When the barrier is opened, the backed up water volume is routed downstream (stage 5). These five stages are computed according to the water level slope in each time step. The methodology to realise the coding of this theoretical concept into a numerical hydrological model is explained in the following chapter 4.

## 4   Methodology to compute backwater effects in rivers and adjacent lowland areas with complex flow control systems

The methodology to calculate backwater effects with a hydrological conceptual approach, consists of three main algorithms: a transfer of discharges to water levels and volumes per stream segment and time step (section 4.1), the calculation of (inter-)

active control structures (section 4.2) and a backwater volume routing according to the water level slope along stream segments and adjacent lowland areas (section 4.3).

## 4.1 Transfer of discharges to water levels and volumes

The flood routing in stream segments of the hydrological network is computed with conceptual hydrological approaches like Kalinin-Miljukov or Muskingum–Cunge (see section 3.1). A transfer of discharges into water levels and volumes is done by
205 calculating the flow regimes using the approaches of Manning-Strickler or Darcy-Weisbach.

According to the Kalinin-Miljukov approach, each stream segment is divided into a cascade of $n$ reservoirs with a characteristic length $L_c$ and the coefficient $K_c$. The WVQ-relations for different states ($n_{wvq}$) in the stream segment are defined with an interpolation between supporting points of water level heights. This results in a division of the bankfull water level height $H_{full}$ (m a.s.l) into ($n_{wvq}$) states with a water level difference $\Delta H$ (m). Three calculation routines are integrated in the
210 flood routing method to compute the flow velocity in stream segments. The appropriate calculation routine is selected according to the stream segment's profile and data availability. Stream segments with a circular profile are computed with the Darcy-Weisbach approach. Stream segments with rectangular or trapezoidal (angular) profiles are computed likewise with the Darcy-Weisbach or with the Manning-Strickler approach. The equivalent sand roughness $k_s$ in (m) using the Darcy-Weisbach approach or the roughness $K_{st}$ (m$^{1/3}$/s) using the Manning-Strickler approach are input parameters. The algorithm of these
215 three calculation routines is illustrated in the flow chart in Fig. 3. The FORTRAN code and equations to compute the following list of flood routing parameters are explained in the *suppl. section 4*: flow velocity $v$, characteristic lengths $L_{km}$, number of characteristic reservoirs $n_{km}$, retention parameters $K_{km}$, water levels $W$, volumes $V$ and discharges $Q$, where $km$ indicates the parameter calculation according to the Kalinin-Miljukov approach.

## 4.2 Calculating (interactive) control functions of drainage systems

A control structure of a linear stream segment is defined with unsteady WVQ-relations and the flood routing is modelled with a storage indication method. In this work the modified Puls method is applied. The outflow of the control structure can be distributed to four receivers (Fig. 4). Operative criteria of control structures are defined for three types of driver time series which are precipitation intensity, water level stages and discharge values. Hydrographs of water level stages and discharges are results given at junction nodes, while precipitation time series are related to subcatchments as spatial input data. The status
of control structures is checked per time step during the execution of the numerical model. A differentiation of control function types is done according to their operative criteria depending on pre-set (external pre-processed), on-the-fly (internal processed) or interactive on-the-fly driver time series. The three control function types and the dependency on the location of the operative criteria are listed in Fig. 4. Control function type (1) depends on observed or externally forecasted driver time series for instance, precipitation intensity or water level gauge data. These control functions are computed in the pre-processing phase
of the simulation run to set the status of a control structure. With forecasted data a time duration can be set to change the status of control functions (closing or opening a gate) with a specific leadtime before the threshold (operative criteria) is reached. In the control functions type (2), criteria depend on the output of computed parameters of the hydrological network, namely water level or discharge. The functions are computed during the simulation run "on-the-fly". This procedure depends on the condition that the driver elements are located upstream of the control structure and are not influenced by backwater. If the criteria of a
control structure depend on downstream or backwater affected conditions in an interactive system, a recursive calculation routine is started to compute the control function type (3). The recursive calculation routine is explained in the following section 4.3.

## 4.3    Calculating backwater effects along river streams and adjacent lowland areas

An afflux due to natural or artificial obstructions (for instance gates or weirs) leads to a rise of water level in upstream segments. To simulate the resulting backwater effects, the downstream directed surplus water volume is reversed as backwater, when the downstream water level is higher than upstream. This concept is illustrated in the theoretical approach in section 3.3 and comprises the simulation of backwater effects, which cause the flooding of upstream lowland areas. The developed algorithm to compute these backwater effects is illustrated in the flow chart in Fig. 5. The calculation routines are nested in computational loops as follows: A spatial loop of streams and areas is nested in a time loop. The time loop is again nested in a backwater system loop.

Each backwater system includes several component types of a hydrological network: linear structures (stream segments), spatial structures (sub-catchments of lowland areas), junction nodes and a control structure (tide gate or water level gauge) at the downstream segment. For the control functions type (1) and type (2) (see section 4.2) the calculation routines (a) to (c) in Fig. 5 are executed while at any element an afflux condition is present (see query: 'Is backwater system active?' = yes). Additionally, per backwater system (j) and per time step (t) a query checks if an interactive backwater system with a control function type (3) is defined. An interactive system depends on both, downstream and upstream conditions. In case of an interactive system, the flag for a 'recalculation' loop is activated. The final balanced stage is reached when in a backwater affected system the downstream water levels are not higher than the upstream water levels within a range of a minimum 'tolerated' water level difference. The method demands to define a minimum difference ($\Delta W_{min}$) according to the application purposes. A smaller tolerated water level difference increases the accuracy of computed water level results. At the same time, this increases the number of backwater computational runs ($k = k + 1$) before reaching a maximum number (currently: $k =$ 10.000). This critical state prevents infinite calculation routines and a warning shows if this limit is reached to check the input parameters, which include an adjustment of the tolerated water level difference. In the exemplary evaluation study (see section 6), a water level difference of about $\Delta W_{min}=0.01$ m gives sufficient results for meso scale stream segments. For local scale stream segments a difference of about $\Delta W_{min}=0.001$ m gives adequate results (Hellmers, 2020). Backwater effects are computed in open stream segments and adjacent lowland areas, which are part of the defined backwater system. For intermediate closed circular profiles having a limited storage capacity, the backwater volume is routed upstream to the next open stream segment.

In the *calculation routine a* (Fig. 5), the initialisation of formal parameters of each linear and spatial data structure for the backwater effect computation is performed. This includes an initialisation of the water level, volume and discharge per time step. Discharges are computed with the flood routing approaches described in section 3.1. The corresponding water levels and retained water volumes are derived from the calculated WVQ-relations per stream segment (see section 4.1). The initialisation of the parameters for the backwater effect computation is illustrated in Fig. 6. For the computation of backwater effects, the formal parameters of each linear and spatial data structure are initialised. This includes an initialisation of the water level, volume and discharge per time step. Discharges are computed with the flood routing approaches described in the supplementary section S3. The corresponding water levels and retained water volumes are derived from the calculated WVQ-relations per stream segment. When the volume in the control structure is increased ($V(t) > V(t-1)$), afflux is generated and the flag for afflux conditions is set to 'true'. The difference in volume between time steps ($\Delta V(t-1)$) is revised continuously during the following backwater calculation routines (b) and (c) (Fig. 5). When the volume in the control system is decreased ($V(t) < V(t-1)$) or not changed ($V(t) = V(t-1)$) the flag for afflux conditions is set to 'false' and the volume ($\Delta V(t-1)$) is reduced by the proportion of the changed volume $\Delta V$ which has been processed already in the time step before. The upstream-directed backwater routing is computed if the 'afflux-conditions-flag' is set to 'true'. The downstream-directed backwater routing is computed if the 'afflux-conditions-flag' is set to 'false'.

In the *calculation routine b* (Fig. 5), the backwater effect computational loop in upstream direction is activated, while afflux conditions are present in the backwater system. The calculation is done per stream segment in a computational loop

starting at the downstream element ($i = n$). If the difference in water levels between the actual and the upstream segment is larger than the defined tolerated water level difference $\Delta W_{min}$, an algorithm to compute the backwater effect is activated. The backwater quantity derived from an afflux at the downstream segment, is routed to the upstream segments. Along the streams, spatial structures (like lowland catchments) are linked, where the water is retained or causes backwater flooding. This developed concept is illustrated in the scheme in Fig. 7, where the backwater effect computation between stream segments with linked spatial structures (retention areas) is shown. The formal parameters of the WVQ-relations of the current (i) and the upstream (i-1) segment are processed. The computation is done in three sub-calculation routines (namely A, B and C) to compute the water level and volume stages.

*Explanation of the sub-calculation routine (A):* In case of adjacent lowland areas (linked spatial data structures), a portion of water flows from the stream segment (i) into the respective linked areas (i) if the water level exceeds the river bank. The inflow continues until the water level in the stream $W_i(t)$ is in balance with the water level in the linked spatial data structures $W_{i,areas}(t)$. The result is a decreased difference in volume $\Delta V_i(t)$ to be routed to the upstream segment ($i-1$) per time step.

*Explanation of the sub-calculation routines (B) and (C):* The computed backwater effect in the calculation routine (B) describes, how the water volume $\Delta V_i(t)$ is added to the upstream linear data structure $V'_{i-1}(t) = V_{i-1}(t) + \Delta V(t)$, whereupon the water level is derived from the WVQ-relations. If the upstream segment is linked with another spatial data structure as illustrated in Fig. 7 (case C), the balancing of water level and volume is done respectively to the procedure in (A). As long as a backwater effect is present in any river segment or adjacent lowland area, the calculation is repeated (till $k = 10\,000$). The algorithm to compute upstream directed backwater effects on the water levels and volumes is illustrated in Fig. 8. If the following queries are true, the upstream backwater effect computation is executed. These queries are called at the beginning of the calculation routine (see: 'Are afflux conditions present?') with the following equation:

$$is\ W_i(t) - W_{i-1}(t) > \Delta W_{min}?\ and\ is\ V_i(t) > V_{i,free}(t)? \tag{1}$$

where the water level $W_i(t)$ (m a.s.l.) and volume $V_i(t)$ (m³) are defined by the WVQ-relation per stream segment with the index i. $\Delta W_{min}(t)$ is the tolerable backwater affected water level rise given for the stream segments (m) in the backwater system. $V_{i,free}(t)$ is the water volume in the segment without backwater effects, which is computed with the flood routing method.

While afflux conditions are present, the water level in the current stream segment (i) is reduced by the minimum water level difference $\Delta W_{min}(t)$. The adjusted storage volume of the stream segment $V'_i(t)$ is defined accordingly by the WVQ-relation. The adjustment of the stream segment (i) is done with the following equations 2 to 4:

$$W'_i(t) = W_i(t) - \Delta W_{min} \tag{2}$$

$$V'_i(t) = f\big(W'_i(t)\big) \rightarrow \text{Derivation of the WVQ-relations} \tag{3}$$

$$\Delta V_i(t) = V_i(t) - V'_i(t) \tag{4}$$

where i' indicates the adjusted stage in the stream segment (i). This results in a difference of volume $\Delta V_i(t)$ which is routed to a linked spatial data structure (for example retention areas). This calculation routine is indicated with (A). Otherwise, the backwater is directly routed to the upstream linear data structure ($i - 1$). These calculation routines are indicated as (B) and (C) in Fig. 8.

In the *calculation routine c* (Fig. 5*)*, the backwater volume is routed downstream, if the afflux conditions at the downstream segment of the backwater system is not present anymore, for instance by opening a gate or starting additional pumping. The water level and storage volume in the stream segments are reduced per time step until free flow conditions are reached. In the developed calculation routine the drainage process of the backed up water volume is calculated. The stream segments are computed in the order from upstream (i = 1) to downstream (i = n). The algorithm for the computation of the subsequently drained backwater in downstream direction is done step wise with the current (i) and the downstream (i+1) data structures using the sub-calculation routines (C) to (A) in reversed order (see Fig. 7 and Fig.8).

In *calculation routine d* (Fig. 5) interactive systems are computed. When a control structure depends on criteria of a downstream backwater affected system, an interactive computational loop is activated. In this case a 'recalculation' loop is started and revises control structure settings if the results of the interactive backwater system are available. Then the recalculation loop restarts the computation of the calculation routines (a) to (c) (Fig. 5). The results of this developed algorithm to compute backwater effects are the time series of water levels (m a.s.l), discharges (m$^3$/s) and volumes (m$^3$) for stream segments and linked spatial data structures (e.g. lowland catchments). Additionally, the activated control functions per control structure are given as time series for verification purposes.

## 5    Implementation of the hydrological method for calculating backwater effects in Kalypso-NA (4.0)

Implementing the developed method into a target software is done for evaluation and application purposes. The implementation is realised in the open source model Kalypso-NA (4.0), which is constantly under development and applied since more than 20 years in research and practice. The numerical model features are: semi-distributed, deterministic, multi-layered and combined conceptual-physical based. The model shows strengths in short computation times, which is in the range of max 3 minutes on typical desktop computers (with e.g. i7-5600U CPU processor) for large catchments (ca. 200 km²) using a time step size of 15 minutes for a 14 days simulation. It is applicable for real-time operational simulations in flood forecasting. In combination with the Kalypso Project providing a user interface, the model Kalypso-NA is applicable for calculating the rainfall-runoff regime in catchments by users, who are not familiar with input scripts. Open access for developments and user application is supported by an online accessible commitment management via Source Forge platform and a wiki as an online manual. More information about the software product Kalypso and the model Kalypso-NA is provided in the *suppl. section 1*. Such an open source module provides the accessibility to the implemented methods and therewith supports to re-use it in other hydrological models. It is the purpose to support a good scientific practice towards open and reproducible science.

The algorithms in the source code Kalypso-NA are extended for the integration of the developed methods for backwater effect computation in rivers and adjacent lowland areas. The hydrological numerical model comprises algorithms in the form of time loops executed within a spatial tree structure (time-before-space algorithm) and spatial calculation routines executed within a time loop (space-before-time algorithm). Both approaches are integrated in the source code of Kalypso-NA (4.0) as illustrated in Fig. 9. A time loop nested in a spatial loop accomplishes the simulation of data structures (such as sub-catchments, stream segments, junction nodes or retention areas) in downstream direction on the basis of the overall results of the upstream data structures. This means that the data structures are computed for the whole simulation period consecutively in the order given by the hydrological network from upstream to downstream. More information about the hydrological network is given in the *suppl. section 3*. The first implementation (Part A) provides actual time-dependent results of data structures to set control functions or drainage criteria in the hydrological network. This method is applied in the extended algorithm to model processes in sub-catchments like the soil water balance and the downstream directed flood routing. This algorithm is explained in more details in the journal paper (Hellmers and Fröhle, 2017).

Additionally, an algorithm is implemented where spatial calculation routines are nested in time loops. This secondary algorithm provides the overall results of a backwater affected system per time step before calculating the next time step. The time loop is additionally nested in a backwater system loop. In that calculation routine the backwater effects in streams and adjacent lowland areas as well as the evaporation from submerged water surfaces are computed. This implementation is labelled as space-before-time algorithm and is illustrated in (Fig. 5). The implemented hydrological model approach is applicable to other catchment studies, while using physical-based input parameters. The input and output parameters are listed in the *suppl. section 2 and 5*.

## 6 Exemplary model application and evaluation

Objective of the model evaluation is to determine the reliability of the numerical model results to be in a sufficient range of accuracy for the designated field of application (Law, 2008; Oberkampf and Roy, 2010; Refsgaard and Henriksen, 2004; Sargent, 2014). An evaluation of the extended model Kalypso-NA (4.0) is performed by comparing the results of the numerical model with observed data of gauging stations in the mesoscale catchment 'Dove-Elbe'. This exemplary catchment comprises a tide gate as well as several sluices, weirs and low lying catchments drained by pumping stations. The drainage through the tide gate depends on low tide conditions. At high tide, the gate is closed causing backwater effects in the streams.

### 6.1 Description of the backwater affected lowland catchment 'Dove-Elbe'

The mesoscale catchment area 'Vier- und Marschlande" has a size of 175 km² and is located in the South-East of Hamburg, Germany (see Fig. 10). The downstream river segment Dove-Elbe is a stream of 18 km in length and a tributary of the tidal influenced Elbe River. Further tributary streams which drain into this main river segment are the Gose Elbe, Schleusengraben, Brookwetterung and a downstream segment of the Bille. These streams are part of the analysed mesoscale catchment. The soil is mainly peat and clay with a varying spatial distribution and thickness. Another regional scale catchment (namely of the river Bille) with a size of about 337 km² drains into the study area 'Vier- und Marschlande'. Thus, an overall catchment area of about 512 km² is drained through the tide gate 'Tatenberger Deichsiel'.

The downstream situated water level in front of the tide gate is affected by a mean tidal range of about 3.7 m (Nehlsen, 2017). The Mean Low Water (MLW) is at about -1.5 m a.s.l. and the Mean High Water (MHW) is at about 2.2 m a.s.l. The tide gate closes when a water level of about 0.9 m a.s.l is exceeded in the Elbe River. During the closure period of the tide gate, water is retained in the stream segments of the 'Vier- und Marschlande' catchment leading to an afflux of water which causes backwater effects. The numerical model includes 75 subcatchments, 75 junction nodes, 75 meso scale stream segments, 7 gauging stations and 7 control structures. These control structures comprise gates, weirs, pumping stations and a tide gate (see Fig. 10). The control functions comprise the opening as well as closure of gates and sluices or starting of pumps according to defined criteria. The backwater affected river segments in the Dove-Elbe with a length of about 12.5 km are characterised with wide profiles (width >100 m) and wide flood prone areas (width >200 m) on the mesoscale.

For the computation of the flood routing, the Kalinin-Miljukov method for mainly irregular profiles with five reservoir parametrisations is applied. An explanation is given in the *suppl. section 4.3*. Additionally, a scenario simulation is performed within the research project StucK (www.stuck-hh.de: Long term drainage management of tide-influenced coastal urban areas with consideration of climate change) with three retention areas (300 000 m²) which are indicated in Fig. 10. The application and evaluation results of the research project StucK for the Dove-Elbe streams as part of the 'Vier- und Marschlande' catchment are summarised in the following section.

### 6.2 Application and evaluation results

An evaluation of the developed method to compute backwater effects with Kalypso-NA (4.0) is done by comparing numerical model results with data of gauge measurements along the river stream segments of the Dove-Elbe. The analysis of two flood events are presented. Measurements of five gauging stations in the Dove-Elbe stream segments are available for a flood event in February 2011 and the measurements of the downstream gauging station are available for a flood event in February 2002. The locations of gauging stations and control structures are indicated in Fig. 10.

The results at the downstream gauging station ("Allermöher Deich") are illustrated in Fig. 11 for the opening and closing function of the tide gate (in red) according to water levels at the downstream gauging station 'Schöpfstelle' in the Elbe River (in dotted violet) for the event in 2002. The tide gate closes when a water level of 0.9 m a.s.l. is exceeded at the downstream gauging station 'Schöpfstelle'. In the illustrated example of February 2002, the tide gate remained closed two times during storm tides. Meaning, the Elbe River water level during low tide periods did not fall below the required minimal water level

of 0.9 m a.s.l. The long closure times generated a large afflux up to a water level of 1.7 m a.s.l. and consequently large backwater effects in the Dove-Elbe streams. The simulated and observed peak water levels show an average difference of about 0.02 m. The differences in peak water levels are in the range of 0.01 m to 0.10 m. This corresponds to a variation of 1 to 10 % in the streams with a backwater affected water level variation larger than 1 m. The Root Mean Square Error (RMSE) ($< 0.12$ m) and coefficient of determination ($R^2$) ($> 0.9$) of the flood event analysis confirm the good result evaluation. The RMSE and $R^2$ show a very good fit for the rising limb of the flood event. Because of an exceptional manual pre-opening of the tide gate by the authority, ca. 1.5 hours before reaching the water level of 0.9 m a.s.l in the Elbe, the simulated control function and observed status of the control structure are not comparable for the falling limb (details are illustrated in *suppl. section 6*). During the rainfall storm event February 2011, the water level increased due to backwater effects caused by high flood discharge from upstream catchments. Here, a difference of less than 0.01 m is shown between observed and simulated peak water levels. The scatter plot, the $R^2$ and the RMSE for the flood event analysis on the 07.02.2011 to 08.02.2011 show a good concordance. An interactive backwater system is present for the downstream Dove-Elbe river section, which is influenced by the control structures 'Reitschleuse' (blue, Fig. 11) and 'Dove-Elbe Schleuse' (green, Fig. 11). Both control structures depend on thresholds of the downstream water levels in the Dove-Elbe stream segments (black, Fig. 11). In this case, the method to model interactive control systems is applied. The evaluation results show a good performance of the model: The closing and opening times of the sluices according to the thresholds are met.

Details and further results of the events February 2002 and February 2011 for the control structures ('Tatenberger Schleuse', 'Reitschleuse' and 'Dove-Elbe Schleuse') are given in the *suppl. section 6*. The average difference in observed and simulated water level peaks is about $\Delta W = 0.04$ m. This corresponds to a difference of about 5 % in relation to the 1 m large fluctuation range of the water table in the stream segments of the Dove-Elbe catchment. Additionally to the good fit in peak values, the hydrographs in the supplement of this article show that the temporal sequence (1) of opening and visa versa closing the control systems and (2) of the rising and visa versa falling limb in the hydrographs in the river segments are well simulated. The results show a good reliability of the computed flood routing and backwater effects in streams. It is stated that with these findings the reliability of the numerical model results are in a sufficient range of accuracy for the designated field of application.

Additionally to the presented evaluation studies, a flood peak reduction measure is analysed in the research project StucK. By excavating three retention areas with a total size of 330.000 m² from +2 m a.s.l. to +1 m a.s.l., an additional retention volume of 330.000 m³ is created when the water level exceeds the river banks at +1 m a.s.l. The location of retention areas is indicated in Fig. 10. With the additional retention volume, the peak water level can be reduced by 0.08 m. For the event 2011 the result is shown in the *suppl. section 6*. More results of the model application for the research project StucK are published in (Fröhle and Hellmers, 2020).

## 7    Discussion of model results and limitations

In low lying lands, backwater effects and backwater induced flooding of areas are an important issue. The literature review in chapter 1 revealed that modelling backwater effects is not or rarely implemented in stand alone hydrological numerical models up to now. In this paragraph, the findings of the presented conceptual method to model backwater effects in lowlands caused by flow control structures using a stand alone hydrological model and the evaluation results are discussed. It points out the achievements and limitations in accomplishing the objectives of this work.

The developed, implemented and evaluated method for modelling backwater effects transfers discharges into water levels using a conceptual approach. Backwater volume routing is calculated by taking into account the water level slope along streams and adjacent lowland areas. The conceptual approach applies a pre-defined water level tolerance to calculate the backwater volume routing. The use of physical-based input parameters (e.g. profile geometries) enables to apply the presented hydrological model for other catchment studies. The input parameters comprise data of the stream profiles, gradients and

roughness along the flow path. The objective in modelling the effects of e.g. tidal ranges on flow control structures and the resulting backwater effects on the flow regime in upstream lowlands are reached by the conceptual hydrological method.

In comparison to coupled hydrodynamic models, the input parameters are parsimonious. Another advantage of the developed method is the direct computation of hydrological processes in backwater affected areas. For example, the infiltration, groundwater recharge, evaporation of water from submerged areas are simulated. To simulate prospected changes in urbanisation or effects by climate change on precipitation patterns can be defined directly in the hydrological numerical model. The implementation of the method is realised in the open source rainfall-runoff model Kalypso-NA 4.0. The conceptual method is re-useable to extend other hydrological models which are based e.g. on the often applied flood routing methods of Kalinin-Miljukov and Muskingum Cunge.

Limitations of the conceptual method exist in modelling details of the spatial and temporal variability in the velocity and tidal flow regime within stream sections. In the conceptual method each stream section is computed as a "reservoir" according to the linear reservoir theory. Meaning, that the backwater profile is assumed to be flat within each river section. The exactness of the water level heights depends on the defined water level tolerance and the scale of the river sections. This means, in contrast to hydrodynamic-numerical approaches, the developed hydrological model does not compute velocity fields within streams and water levels represent average values per stream segment. This hydrological flood routing method is appropriate to accomplish the objectives of this work to model regional scale backwater affected catchments (>100 km²) with the requirement to keep the computing times small and with a parsimonious parametrisation. It does not replace the demand to model two or three-dimensional velocity fields and to compute the distribution of water levels within streams or submerged areas by the use of coupled hydrodynamic-numerical models for specific research questions.

The evaluation results (chapter 6) show the applicability of the model for simulating rainfall-runoff regimes and backwater effects in an exemplary lowland catchment (175 km², Hamburg, Germany) with a complex flow control system and where the drainage is influenced by a tidal range of about 4 m. The flood event analysis confirm good evaluation results: the comparison of observed to simulated results show a low RMSE ( < 0.12 m) and a high R² ( > 0.9). In the presented application studies a standard desktop computer with i7-5600U CPU processor and 2.6 GHz is applied. The computation time is in the range of max 3 minutes even for large catchments (here 175 km²) using a time step size of 15 minutes for a 14 days simulation. With these short simulation times the presented method shows a good potential to be used in flood forecast simulation models, where results in form of time series (e.g. water level and discharge) per river section and flood prone area are sufficient.

## 8    Summary and outlook

Numerical models are required in forecast simulations and to assess the consequences by future impacts like changes in magnitude as well as probability of stormwater events, changes in urbanisation and predicted mean sea level rise on the runoff regime in catchments. Especially in coastal lowlands, the pressure on stormwater drainage and flow control systems raises due to a combination of all three impacts. The literature review shows weaknesses in modelling water depths and backwater effects in streams and lowland areas using stand alone hydrological numerical models. A method to resolve these weaknesses is presented in this article. The developed numerical method is:

  (1)  applicable to model complex drainage and flow control systems in backwater affected lowlands,

  (2)  efficient by using short runtimes for real-time operational model application,

  (3)  open for further model developments,

  (4)  re-useable for other hydrological model solutions and

  (5)  parsimonious with respect to the complexity of input parameters.

The evaluation results in the application study of the complex and tidal influenced lowland catchment 'Vier- und Marschlande' illustrate good conformance in the simulated backwater effects on the flow regime. Additionally to the findings in this article,

the published outcomes in (Hellmers, 2020; Fröhle and Hellmers, 2020) show the reliability of the numerical model results to be in a sufficient range of accuracy for the designated field of application to answer a wide range of hydrological and water management questions. The numerical model is suitable for operational flood forecasting, real-time control, risk analyses, scenario analyses and time series gap filling in micro to regional scale catchments. The presented method is re-useable for other hydrological numerical models which apply conceptual hydrological flood routing approaches (e.g. Muskingum-Cunge

or Kalinin-Miljukov).

**Outlook**

The presented method in the model Kalypso-NA (4.0) to compute backwater affected flood routing will be adapted to model hydrological processes in local scale drainage measures (aka SUDS, GI, BMP as parts of nature based solutions). Preliminary

research study results of local scale drainage measures are published in (Hellmers and Fröhle, 2017) and in (Hellmers, 2020). The integration of Kalypso-NA in flood forecasting systems (e.g. Delft-FEWS) is in progress.

## 9    Acknowledgement

The model development and evaluation study is part of the research project StucK (Long term drainage management of tide-influenced coastal urban areas with consideration of climate change; 2015-2019; www.stuck-hh.de). The joint project in the

500 funding measure 'Regional Water Resources Management for Sustainable Protection of Waters in Germany'' (ReWaM) is sponsored by the German Federal Ministry of Education and Research (BMBF). Publishing fees are supported by the Funding Programme 'Open Access Publishing' of Hamburg University of Technology (TUHH). The authors gratefully acknowledge this support.

## 10    Software availability.

*Name of the modified computation model*: Kalypso-NA (version 4.0)

*Developer of the modified part:* (IWB) Institute of River and Coastal Engineering (TUHH-Hamburg University of Technology)

*Contact address*: Denickestrasse 22, 21073 Hamburg, Germany.

*Phone*: +49 4042878 4412.

*Homepage*: https://www.tuhh.de/wb/forschung/software-entwicklung/kalypso/kalypso-na.html

*First time available*: BCENA renamed to Kalypso-NA (around 2000).

*License*: GNU Lesser General Public License (LGPL) as published by the Free Software Foundation, version 2.1.

*Hardware required*: PC

*Program language*: FORTRAN

*Program size*: 5.8 MB

*Availability and cost:* Compiled code is freely available at http://kalypso.wb.tu-harburg.de/downloads/KalypsoNA/. Source code of the modified part of the model presented in this paper is published in (Hellmers, 2021) (DOI: 10.15480/882.3522; http://hdl.handle.net/11420/9508). Main code sections of flow diagrams and equations are published in the supplement of this article.

## 11    Author contribution

The lead author of this article, SH, formulated the research topic. She placed the topic in the current state of research and defined the purpose of the work. The presented approaches, methods, implementations and evaluation results have been

worked out by SH and were discussed with PF. The conceptualization of the paper was a joint effort from SH and PF, as were the discussion and refinement of the methods presented.

## 12 Competing interests:
The authors declare that they have no conflict of interest.

## 13 Review statement

This paper was edited by Jeffrey Neal and reviewed by two anonymous referees.

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

## 15 Figures

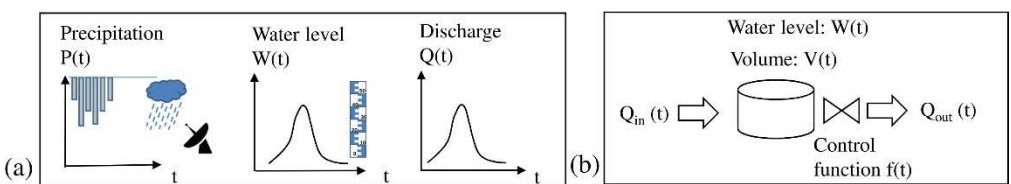

**Figure 1: (a) Illustration of operative criteria in a control function depending on driver time series of precipitation, water level and**
615 **discharge. (b) Scheme of a control structure with a control function changing the water level W(t), volume V(t) or outflow Q(t) per time step t.**

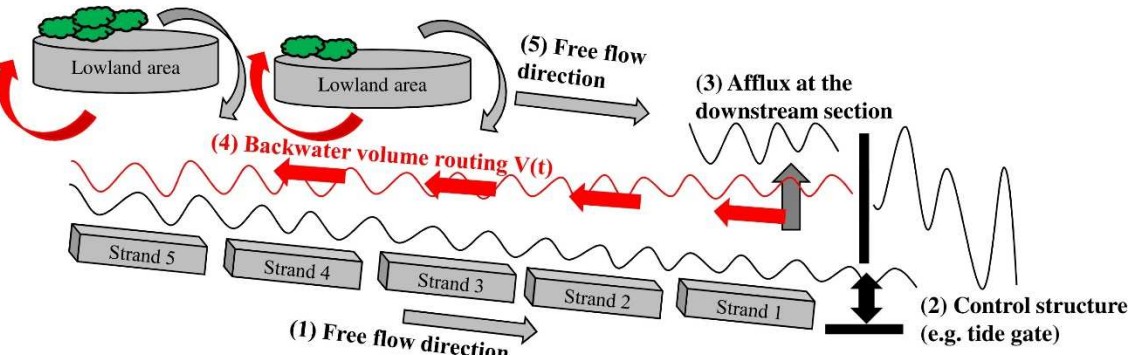

**Figure 2: Scheme of five computation steps in the developed concept to compute backwater effects with a hydrological approach:**
620 **(1) free flood routing computation downstream, (2) control structure simulation, (3) afflux computation, (4) backwater volume routing computation in upstream direction including adjacent lowland areas (as well as retention areas) and (5) free flood routing computation after opening the barrier.**

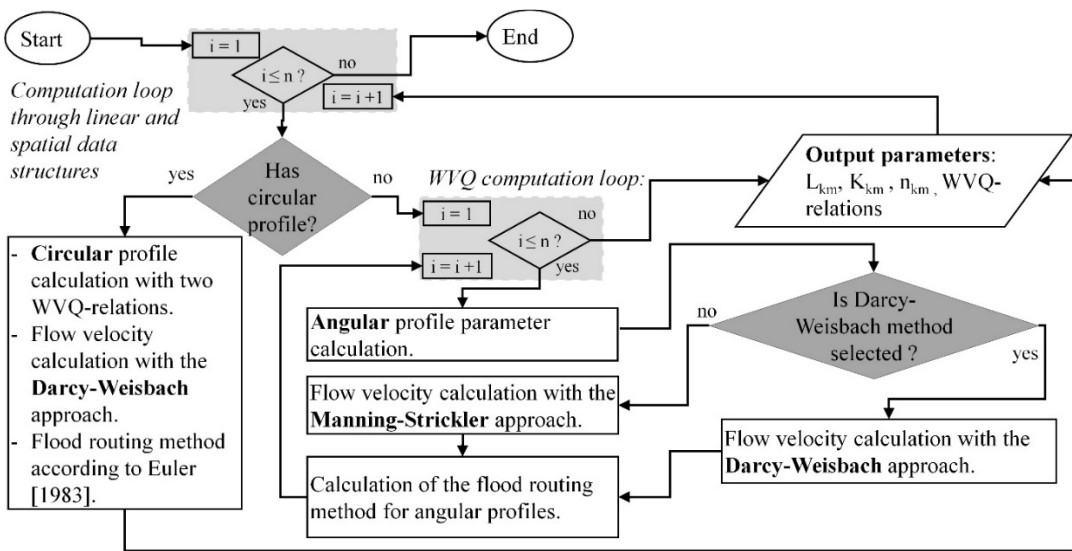

**Figure 3: Algorithm to compute the relations between water level, volume and discharge (WVQ) per stream segment.**

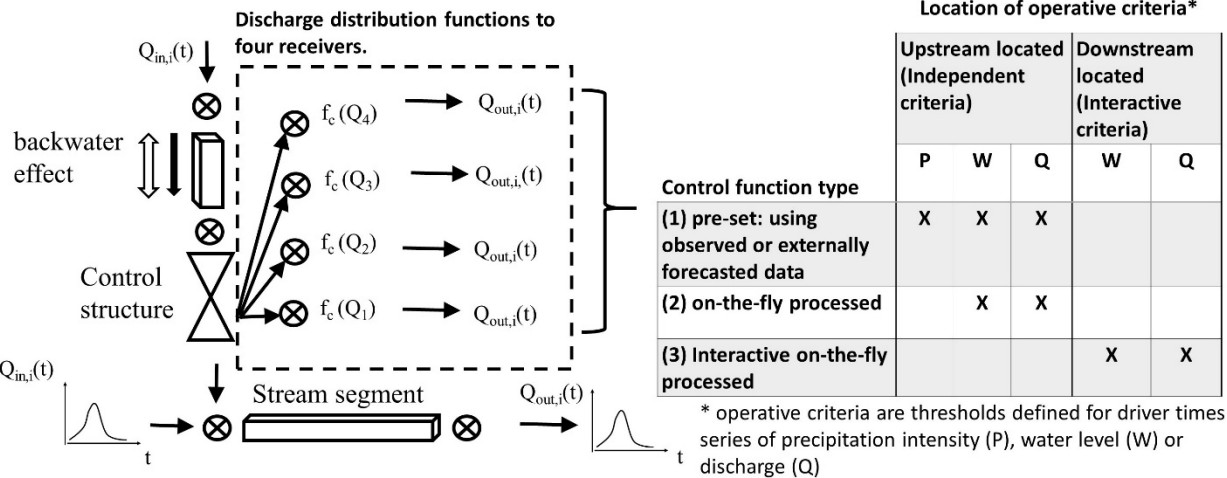

| Control function type | Location of operative criteria* | | | | |
|---|---|---|---|---|---|
| | Upstream located (Independent criteria) | | | Downstream located (Interactive criteria) | |
| | P | W | Q | W | Q |
| (1) pre-set: using observed or externally forecasted data | X | X | X | | |
| (2) on-the-fly processed | | X | X | | |
| (3) Interactive on-the-fly processed | | | | X | X |

\* operative criteria are thresholds defined for driver times series of precipitation intensity (P), water level (W) or discharge (Q)

**Figure 4: Scheme of a control structure with discharge distribution functions to four receivers and the three control function types depending on operative criteria.**

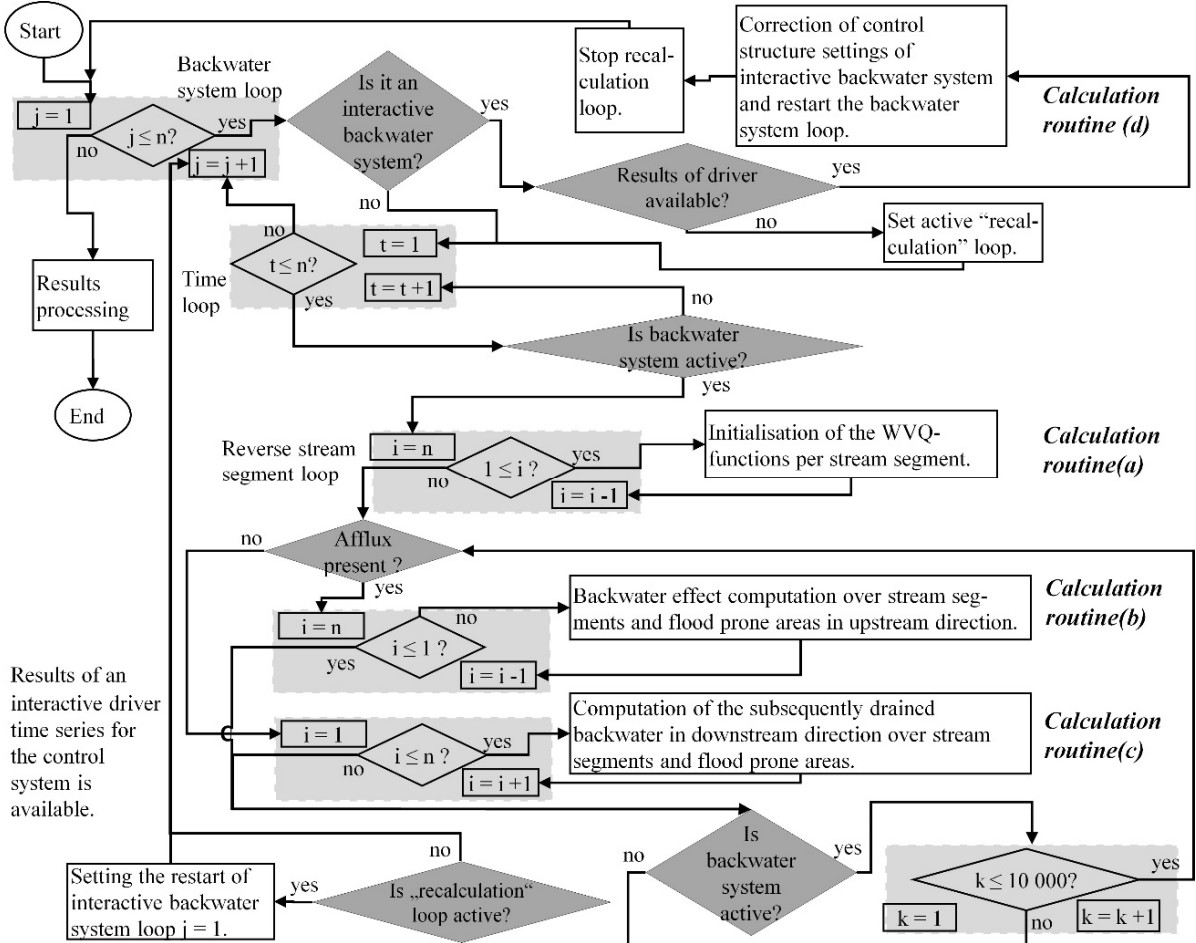

**Figure 5: Algorithm to compute backwater effects in streams and lowland areas (like retention areas) with the indicated calculation routines (a, b, c, d). It is realised with a space-before-time algorithm for modelling backwater effects and control structures per backwater system.**

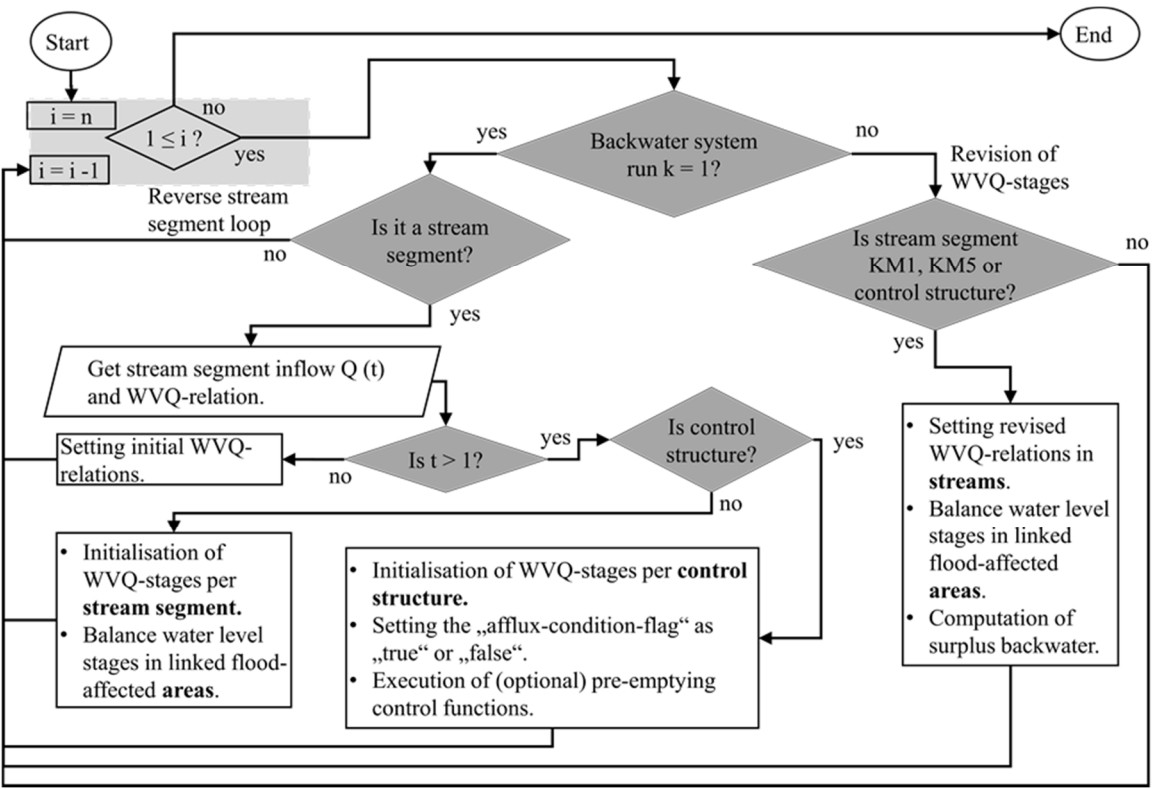

**Figure 6: Algorithm to initialise WVQ-relations in streams, control structures and areas per backwater system. Illustrated details of the first calculation routine (a) in the algorithm in Fig. 5.**

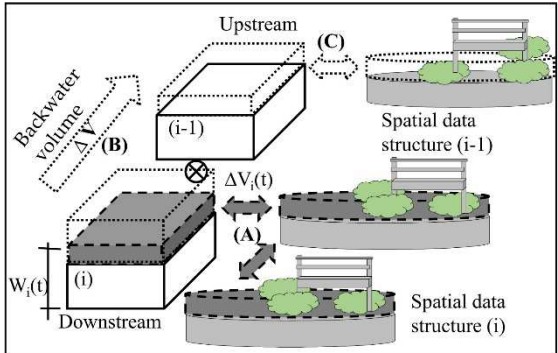

**Figure 7: Scheme of the sub-calculation routines (A), (B) and (C) to compute backwater effects in stream segments and adjacent lowland areas (spatial data structures). The sub-calculation routines are part of the main *calculation routine b and c* (Figure 5).**

```
Start → Reverse stream segment loop → i = n → 1 ≤ i ? — yes → End
                                                  no
                                          i = i - 1
```

Are afflux conditions present? — yes → Computation of the adjusted water level $W_i(t)$ and volume $V_i(t)$ in stream segment (i).

Linked spatial structures are present? — yes

Backwater volume computation $\Delta V_i(t)$.

no

**(B)** Backwater effect computation between streams (i) and (i-1).

**(C)** Backwater effect computation between upstream segment and linked areas (i-1).

Adjusted backwater volume computation $\Delta V_i{}'(t)$.

**(A)** Backwater effect computation between current stream segment and linked areas (i) (e.g. retention areas). With an iteration loop if multiple areas are linked.

**Output:** V(t), W(t) of current (i) and upstream (i-1) segment and linked areas.

**Figure 8: Computation of the upstream directed backwater effect over stream segments and adjacent lowland areas (e.g. retention areas). Illustrated details of the second calculation routine (b) of the algorithm in Fig. 5.**

```
Start → Primary time-before-space algorithm:
        • Computation of hydrological processes.
        • Modelling the flood routing in downstream direction.
      → Secondary space-before-time algorithm :
        • Computation of backwater effects.
        • Modelling interactive control structures.
      → Output of results → End
```

**Figure 9: Structure of the implemented primary and secondary algorithm in the source code of Kalypso-NA (4.0). The enhancement of the primary algorithm is published in (Hellmers and Fröhle, 2017). The new (secondary) algorithm is explained in section 4.3.**

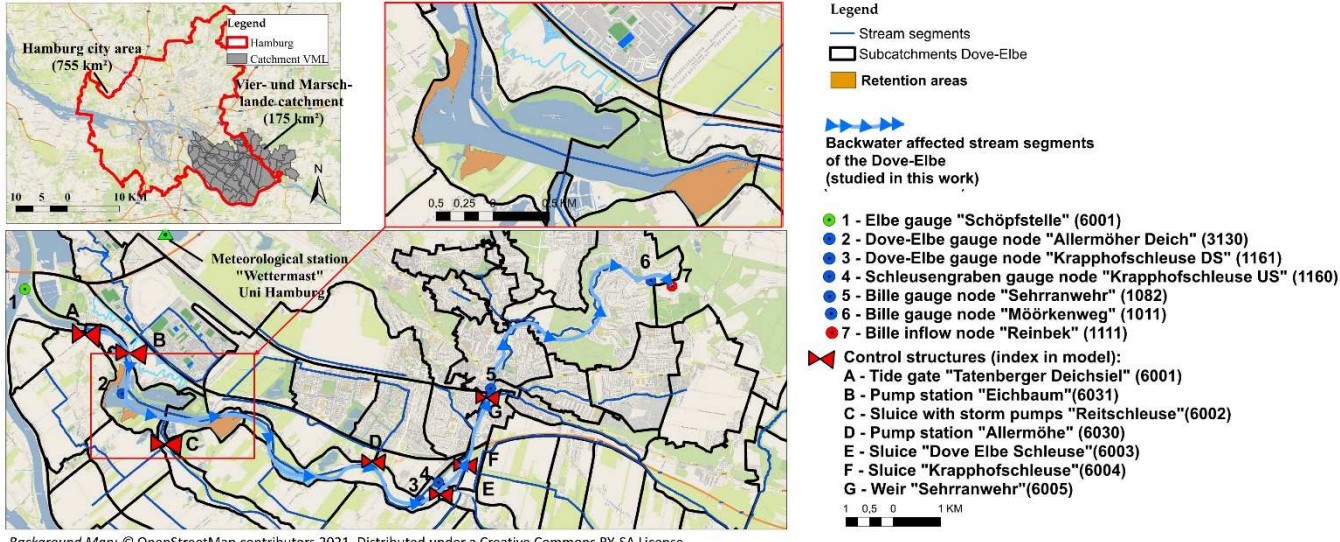

**Figure 10: Map of the application study area 'Vier- und Marschlande' (175 km²): subcatchments, gauging stations (1 to 7), studied backwater affected streams of the Dove-Elbe, three retention areas in the main stream and control structures (A to G).**

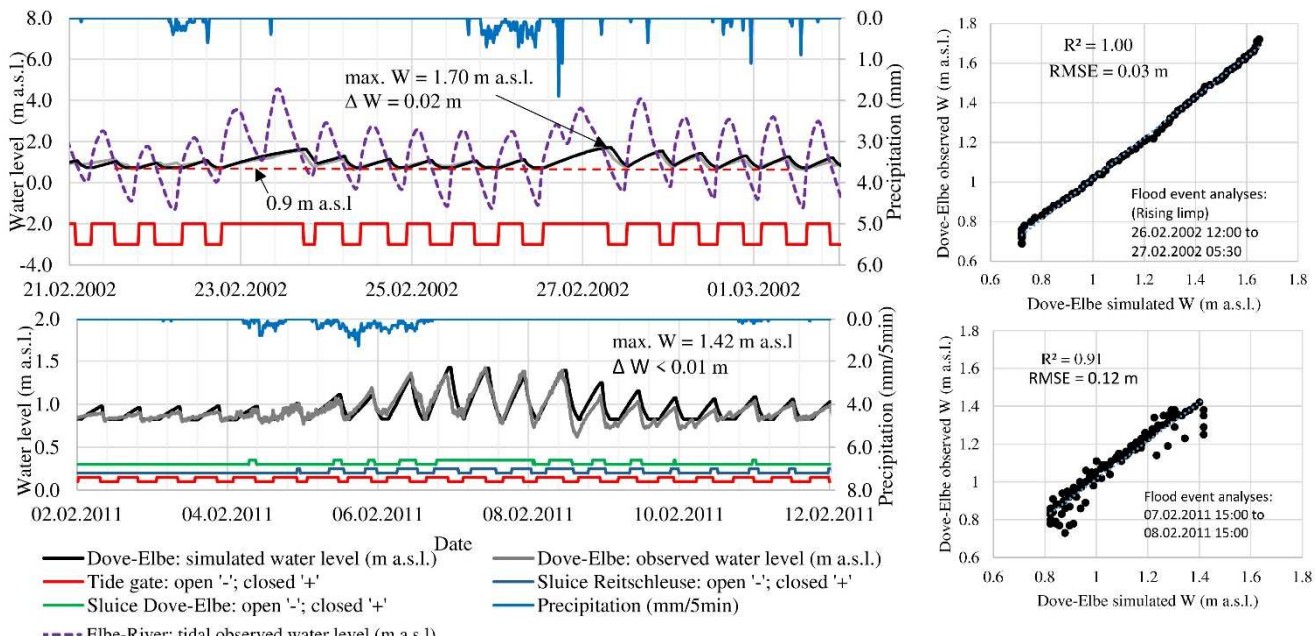

**Figure 11: Closure and opening state of the control structures as well as simulated and observed water levels at the downstream gauge 'Allermöher Deich' for the event February 2002 and February 2011. The tide gate remained closed two times during the storm events in February 2002. Meaning, the Elbe River water level during low tide period did not fall below the required minimal water level of 0.9 m a.s.l. The simulated and observed water levels depict a difference of 0.02 m to 0.01 m in a stream with a water table fluctuation of about 1 m. The RMSE for the flood event analysis shows a deviation of up to 0.12 m.**
