# Peer review of "Computation of backwater effects in surface waters of lowland catchments including control structures – An efficient and re-usable method implemented in the hydrological open source model Kalypso-NA (4.0)"

_Geoscientific Model Development, 2021_

## Referee Comment (RC1)

**Review of the paper from Hellmer & Fröhle**

**Common remarks**

The contribution is in line with the aims and scope of GMD. The paper represent a sufficiently substantial advance in modelling science? The consideration of backwater effects like realized is a novel aspect of modelling. The methods and assumptions are valid and clearly outlined. Nevertheles some additional remarks could be useful (see detailed comments). Alltogehter a minor revision is recommended.

The structure of the paper and the figures are appropiate. Partly the figures have a high information density. Therefore it is difficult to recognize all details. But it is due to the nature of the subject.

**Remarks about content-related aspects**

Line

| | |
|---|---|
| 28 | I would say that (1) is part of (2) |
| 116 /117 & 120 / 121 | this words are the identic, perhaps both models can be evaluated togehther |
| 171 | such structures control more the local and regional water levels than the flow of whole catchments, the influence on the discharge is rather short-term after operations |
| 221 / 222 | precipitation as part of subcatchments sounds a little bit strange, perhaps the following is better: "while precipitation time series are related to subcatchments as spatial units" |
| 274 | "changed differences" sounds not clear enough, would be the words" decreased volume" better? |
| 386 | compared to other passages the results are not discussed here |
| 421 – 425 | this are detailed results and not usual in a summary, partly they are a repeat |

**Discussion of 4.2:** Three functions are dicussed and their operative criteria are mentioned (Line 223). If the reviewer has not overlooked anything, than is not clear what this criteria are and and how they are used to choose one of the three functions? It it is true, some additions would be useful. Besides in figure 4 on the left side 4 functions with Q1 to Q4 are listed. What is their meaning compared to the 3 function on the right side. A explanation is *function . . ., but there is not an additional star * in the picture.

**Line 269 / 270:** If the reviewer has not overlooked a special remark, than it is not discussed how the retention quantity is calculated. Perhaps GIS is used or similar?

**Finally:** In the text I have found some remarks which are repeats of remarks in other chapters, for example line 243 to 245. Therefore the impression is, that curtailments are possible. But it is not mandatory.

**English language**

The reviewer is not expert for English. Possibly the following recommendations could be useful:

Line

8 / 9        "constrol structures" instead of "drainage structures"

16           "simulating" for "modeling"

41           "will be faced by higher pressures" for "will face"

62 / 63      "impact on flow regime"

64           "outlook on" for "outlook of"

89           "like by the frequently used"

314          is "are extendable" better?

317          are the words "integrated as extensions" better suited?

321          "given by" instead of "given in"

344          the second "is" is not necessary

384          "concordance" instead of "result"

415          "The use of" instead of "Using"

**Editorial corrections**

96           cancel ":" after models

164          "change" for "changes"

206          perhaps n should be used already here: between n supporting
             points

225          "depends" for "depend"

Some passages are very long wherefore a subdivision is recommended, for example: beginning with line 245 or line 343 to line 363 (21 lines)

---

## Author Comment (AC3)

29.11.2021

Answer R1:

Thank you very much for the detailed and well worked out review. Your comments, questions and remarks are very valuable for our revision of the preprint. We are pleased to discuss your points of the review in more details to make our paper and statements more clear. For that purpose we cite your comments in the review (with _"", italic, underlined_) and answer each comment thereafter.

- Reviewer Comment 1:

  _"Common remarks_

  _The contribution is in line with the aims and scope of GMD. The paper represent a sufficiently substantial advance in modelling science? The consideration of backwater effects like realized is a novel aspect of modelling. The methods and assumptions are valid and clearly outlined. Nevertheles some additional remarks could be useful (see detailed comments). Alltogehter a minor revision is recommended. The structure of the paper and the figures are appropiate. Partly the figures have a high information density. Therefore it is difficult to recognize all details. But it is due to the nature of the subject. "_

  Answer to comment 1:

  Thank you for your positive comment. We are pleased to tell you that we will add another chapter about a more nuanced discussion of the applicability and limitation of our proposed conceptual method.

- Reviewer Comments 2: "_Remarks about content-related aspects:_"

- _"Line 28  I would say that (1) is part of (2)"_

  Answer: Thank you for the remark. We distinguish here between the (1) compartment of the surface-atmosphere interaction and (2) the compartment of the soil-vegetation-atmosphere. In (1) the evaporation and in (2) for example, the transpiration is regarded. Because of a differentiation if vegetation is present or not, we think it is reasonable to distinguish between these compartments.

- _"116 /117 & 120 / 121 this words are the identic, perhaps both models can be evaluated togehther"_
  Answer: Thank you for the remark. We agree in merging the sentences as follows:

  "The hydrological model 'ArcEGMO' takes into account backwater effects by hindering the downstream flood routing when the water level at the downstream segment is higher than the upstream one (Pfützner, 2018).

 The method presented by National Hydrological Forecasting Service in Hungary (Szilagyi and Laurinyecz, 2014) applies a discrete linear cascade model to account for backwater effects in flood routing by adjusting a storage coefficient of the cascade. The ArcEGMO and NHFS method calculate a retained flood rooting, but neither computes backwater volume being routed into upstream segments by a reverse flow direction nor the backwater induced flooding of adjacent lowland areas."

- *"171 such structures control more the local and regional water levels than the flow of whole catchments, the influence on the discharge is rather short-term after operations "*

  Answer:
  Thank you for the remark. We agree to describe the effect of these control structures more precisely. Revised sentence: "Backwater effects in river sections are often caused at obstacles like weirs, (tide) gates, retention or detention reservoirs, which also function as control structures in streams".

- *"221 / 222 precipitation as part of subcatchments sounds a little bit strange, perhaps the following is better: "while precipitation time series are related to subcatchments as spatial units"*

  Answer:

  Thank you for the remark and suggestion. We revised the sentence in the following way: "Operative criteria of control structures are defined for three types of driver time series which are precipitation intensity, water level stages and discharge values. Hydrographs of water level stages and discharges are results given at junction nodes, while precipitation time series are related to subcatchments as spatial input data."

- *"274 "changed differences" sounds not clear enough, would be the words" decreased volume" better?"*

  Answer: Thank you. We agree in this change.

- *"386 compared to other passages the results are not discussed here"*

  Answer:

  The paragraph is as follows: "An interactive backwater system is present for the control structures 'Reitschleuse' (blue, Fig. 11) and 'Dove-Elbe Schleuse' (green, Fig. 11) which depend on thresholds of the downstream water levels in the Dove-Elbe stream segments (black, Fig. 11). In this case, the method to model interactive control systems is applied and evaluated."

  We agree in completing the sentence with a statement about the evaluation result:

  "An interactive backwater system is present for the downstream Dove-Elbe river section, which is influenced by the control structures 'Reitschleuse' (blue, Fig. 11) and 'Dove-Elbe Schleuse' (green, Fig. 11). Both control structures depend on thresholds of the downstream water levels in the Dove-Elbe stream segments (black, Fig. 11). In this case, the method to model interactive

control systems is applied. The evaluation results show a good performance of the model: The closing and opening times of the sluices according to the thresholds are met."

- _"421 – 425  this are detailed results and not usual in a summary, partly they are a repeat Discussion of 4.2: Three functions are dicussed and their operative criteria are mentioned (Line 223). If the reviewer has not overlooked anything, than is not clear what this criteria are and and how they are used to choose one of the three functions? It it is true, some additions would be useful. Besides in figure 4 on the left side 4 functions with Q1 to Q4 are listed. What is their meaning compared to the 3 function on the right side. A explanation is *function . . ., but there is not an additional star * in the picture."_

Answer: Thank you for the comment. The text is as follows: "The differences in peak water levels are in the range of 0.01 m to 0.10 m. This corresponds to a variation of 1 to 10 % in the streams with a backwater affected water level variation larger than 1 m. The RMSE ( < 0.12 m) and $R^2$ ( > 0.9) of the flood event analysis confirm the good result evaluation." We agree to reduce the details in the summary and give a reference to the results in paragraph 6.2 (not 4.2 as mentioned by the reviewer).

To paragraph 4.2: The question of the reviewer refers to the following text: "The status of control structures is checked per time step during the execution of the numerical model. A differentiation between three functions of control structures is done according to their operative criteria depending on pre-set (external pre-processed) or on-the-fly (internal processed) driver time series. The three functions of control structures and operative criteria are listed in Fig. 4."

Answer: We agree to describe the functions and criteria in a more nuanced way. The figure 4 and the text will be revised.

- _"Line 269 / 270: If the reviewer has not overlooked a special remark, than it is not discussed how the retention quantity is calculated. Perhaps GIS is used or similar? "_

Answer: Thank you for the question. The sentence is: "The backwater quantity derived from an afflux at the downstream segment, is routed to the upstream segments." The "routing" of backwater in upstream direction is calculated in a simplified way not taking into account the roughness up to now. Taking into account the roughness parameters in a conceptual way is an outlook of the proposed method.

- _"Finally: In the text I have found some remarks which are repeats of remarks in other chapters, for example line 243 to 245. Therefore the impression is, that curtailments are possible. But it is not mandatory._

Answer: Thank you for the comment. We agree in this and revised the sentences.

_"English language: The reviewer is not expert for English. Possibly the following recommendations could be useful:_

- _Line 8 / 9  "constrol structures" instead of "drainage structures"_ (Answer: We agree! Thank you!)

- *16 "simulating" for "modeling"* (Answer: We agree! Thank you!)
- *41 "will be faced by higher pressures" for "will face"* (Answer: We agree! Thank you!)
- *62 / 63 "impact on flow regime"* (Answer: We agree! Thank you!)
- *64 "outlook on" for "outlook of"* (Answer: We agree! Thank you!)
- *89 "like by the frequently used"* (Answer: We agree! Thank you!)
- *314 is "are extendable" better?* (Answer: Don't agree. We extended the code already.)
- *317 are the words "integrated as extensions" better suited?* (Answer: Thank you! The sentence is shortened as follows: "Both approaches are integrated in the source code of Kalypso-NA (4.0) as illustrated in Fig. 9."
- *321 "given by" instead of "given in"* (Answer: We agree! Thank you!)
- *344 the second "is" is not necessary* (Answer: We agree! Thank you!)
- *384 "concordance" instead of "result"* (Answer: We agree! Thank you!)
- *415 "The use of" instead of "Using"* (Answer: We agree! Thank you!)
- *Editorial corrections*
- *96 cancel ":" after models* (Answer: We agree! Thank you!)
- *164 "change" for "changes"* (Answer: We agree! Thank you!)
- *206 perhaps n should be used already here: between n supporting points* (Answer: We agree! Thank you!)
- *225 "depends" for "depend"* (Answer: We agree! Thank you!)

- *Some passages are very long wherefore a subdivision is recommended, for example: beginning with line 245 or line 343 to line 363 (21 lines)*

  (Answer: Thank you for the comment! We will take a separation of paragraphs into account during the revision of the preprint.

---

## Author Comment (AC4)

29. November 2021

Answer R2:

Thank you very much for the detailed and well worked out review. Your comments, questions and remarks are very valuable for our revision of the preprint. We are pleased to discuss your points of the review in more details to make our paper and statements more clear. For that purpose we cite your comments in the review (with *"", italic, underlined*) and answer each comment thereafter. The specific comments are answered first, before responding to the overall comments.

- Reviewer Comment 1:

*"Line 23-24: "Open demand exists in hydrological modelling of rainfall-runoff regimes in lowlands which are distinguished by complex flow routing in mostly intensively drained catchments by manifold control structures." I think this sentence tries to say too many thigs, consider splitting up the points being made."*

Answer to comment 1:

Thank you for the comment. We are pleased to split up this sentence as follows: (Line 23-24 ) "Open demand exists in hydrological modelling of rainfall-runoff regimes in lowlands. The flow routing in lowland catchments is characterised by artificially drained catchments using manifold control structures."

- Reviewer Comment 2:

*"Introduction. Traditionally backwater and inundation process would be simulated by a coupled hydrodynamic model (of which many are available). I think this needs to be discussed and then a clear reason for including such processes within the hydrological model can be set out. At the moment the introduction only discussed modelling of rainfall runoff as an isolated field of research. As a reader I immediately ask why not couple to another model. I appreciate that this is visited later in the manuscript."*

Answer to comment 2:

Thank you for the comment and advice. We agree in your point of view and your arguments. We are looking especially from the hydrological model point of view on this topic. Improving the functionality of a hydrological conceptual model is our objective in this paper. Our proposed conceptual method doesn't intend to substitute the application of hydrodynamic numerical models for computing, for example, flood inundation maps. Therefore, we agree in explaining the intention of the proposed hydrological conceptual method in detail already in the introduction. The discussion of applying hydrodynamic models and/or hydrological models is shifted from line 73 – 84 to the introduction in line 36 ff. In this way, the reader is informed earlier about the intention of the proposed hydrological method. Additionally, the limitations of our proposed conceptual method will be explained in a new additional chapter in the paper. In this way a more nuanced conclusion will be given at the end about the applicability and the limitation of the proposed method.

The text of the shifted lines in the introduction (36- 49):

"Simulating backwater effects, velocity fields and the spatial distribution of water depths for flood inundation maps demands for 2D or 3D hydrodynamic-numerical models with the numerical integration of the partial differential equations describing the flood routing processes. To compute spatial detailed simulation results in river streams and flood plains, coupled
hydrological and hydrodynamical model approaches fit well to meet the required modelling objectives. But, hydrodynamic-numerical models require larger effort to parameterise river streams and simulation times, which are at least one to two orders of magnitudes longer in comparison to conceptual hydrological flood routing approaches to model river streams. High resolution data describing the topography of the main channel and the natural flood plain in the
case of bank overflow is necessary. Hence, the availability of suitable detailed profile data from measurements is significant for hydrodynamic-numerical modelling. The larger effort in data resources and runtime for hydrodynamic-numerical model simulations is no limitation for answering special research questions and to create detailed inundation maps. However, applying a coupled hydrological-hydrodynamic model shows disadvantages in the application on meso to
regional catchment scales (>100 km²) and for operational forecast applications. Therefore, it is proposed in this article, that a stand alone hydrological approach can be beneficial in flood forecasting models to enable parsimonious and efficient modelling of flood routing and backwater effects in lowlands, by a conceptual hydrological method producing less detailed results."

• Reviewer Comment 3:

*"There is some rather vague language used in places that detracts from the writing. For example, on line 46 "new concepts are required." What are new concepts? And then with regard to "this article fulfils five objectives in hydrological modelling" it would be more normal to set out the four objectives and then discuss the success of meeting them after the*
*results have been presented."*

Answer to comment 3:

Thank you for your remark. We agree in substantiate the sentence in line 46 and agree to revise the structure of the text. In the revised structure, the objectives are described first to meet the revealed shortcomings in hydrological modelling. Additionally, we discus the
met objectives and point out the limitations in more detail in a new chapter (7 discussion of results).

• Reviewer Comment 4:

*"Line 55 "Most promising to accomplish the defined five objectives for a re-usable, open,*
*efficient and parsimonious hydrological model, is the development of an extension approach for state-of-the-art flood routing methods (for instance Muskingum-Cunge or Kalinin-Miljukov), which can be transferred and implemented in different hydrological numerical model approaches and on different model scales." Could this be more specific to your study objective, which I think are to have a scheme that can simulate the*
*backwater effect of river and floodplain flows. This might just be my take on it but the objectives seem broader than those set out in the abstract and title."*

Answer to comment 4:

We appreciate your remark about this sentence and agree in your comment that the describtion of objectives require a revision. This goes along with the answer to comment 3. The sentence is revised as follows: "To accomplish the defined five objectives for a re-usable, open, efficient and parsimonious hydrological method to model backwater effects, the authors suggest to develop a conceptual extension approach for state-of-the-art flood routing methods (for instance Muskingum-Cunge or Kalinin-Miljukov)."

- Reviewer Comment 5:

*"Line 67-69. These statements could do with some references."*

Answer to comment 5:

Agree. References are added.

- Reviewer Comment 6:

*"Line 76 "(2) future impacts of climate change and urbanisation are not directly parameterised in the model approach" I don't agree with this statement. They are included to the extent that they are included in whatever forcing is coming from the models boundary conditions. It's also common to adjust friction values in such model or edit the topography to explicitly represent urbanisation – if anything urbanisation is more explicitly represented in a hydrodynamic model than what its being compared with. I agree with points 3 and mostly with point 1 - although there are examples of hydrodynamic models being applied in quite data scare settings with limited parametrisation and topographic data."*

Answer to comment 6:

Thank you for the remark and discussion. Our intention is to apply parameters of landuse maps like the sealing rate of partially impermeable surfaces and parameters of the spatial distribution of vegetation types (root depth, LAI) and spatially distributed rainfall data series as input. When considering stand alone hydrodynamic models, we agree in the argumentation that indirect parameters are derived (e.g. friction values) to represent the impact of urbanisation. We see here a dependancy on coupling the hydrodynamical model to a hydrological model for representing the catchment characteristics. We agree, that this argumentation remains vague and that it can not be given without further explanations. Therefore it is not given in this context anymore.

- Reviewer Comment 7:

*"Line 100: I didn't understand the use of the word 'decisive'. Furthermore, the rest of the sentence lacked context for me."*

Answer to comment 7:

Thank you, we agree in revising the sentence. "In (Waseem et al., 2020), a review of models is published with regard to simulate important hydrological processes in coastal lowlands. This review shows weaknesses in the model SWIM (soil and water integrated model) and HSPF (hydrological simulation program—FORTRAN). The approaches in the models SWAT (soil and water assessment tool) und MIKE SHE show good conformity to simulate processes in lowlands while both are not applicable to model backwater effects in the river, on floodplains or other adjacent lowlands and backwater effects caused by control structures (sluices, pumping stations and tide gates).

- Reviewer Comment 8:

*"Section 4: I found the method difficult to follow because it is split over several sections*
*and the supplement. If I understand correctly when the downstream level exceeds an upstream level volumes of water are moved to the upstream cell in increments of Wmin until the excess height downstream is less than Wmin? Water can be further rooted onto floodplain storage (linked areas) via the same method in a sub loop. If this is wrong then I haven't understood the method! "*

Answer to comment 8 (Part 1):

Your description of the conceptual method is fine.

*"Section 6 seems quite critical to the method to me so it is a bit odd that its not in the main text, but I'm happy to listen to justifications of why this should be in the supplement. What I think is missing here is a description of the hydraulic assumptions being made and how*
*these might differ from reality. I think the main assumption is that the backwater profile is flat (termed "final balanced stage" in the text I think) and what this means is that as the water level downstream increases the components upstream progressively become part of the same flat pond or bathtub. How does this differ from the hydrodynamic backwater effect? Does this mean that any tidal signal will be instantaneously routed upstream*
*rather than propagating like a wave? I think this is fundamental to any discussion around general applicability."*

Answer to comment 8 (Part 2):

Thank you for your opinion. Our purpose was to simplify the reading flow of the main text and thought that the mathematical description of the method interrupts the flow of
reading. After your comment we see that providing the mathematical details right away in the text is more important. We will move the mathematical details of section 5 and 6 of the supplementary back into the main text. The text is added were the references were given.

- Reviewer Comment 9:

*"The method seems pragmatic and sensible to me, but I'm not sure I fully appreciate the assumptions and limitations relative to a shallow water wave simulation and where this method might become inaccurate. Could be added?"*

Answer to comment 9:

We agree in pointing out the limitations and assumptions of the proposed conceptual method in more detail. This text will be given in an additional chapter (7) and the text in the summary (chapter 8) will be reduced.

*"Line 333 "The compiled code is freely available at http://kalypso.wb.tu-harburg.de/downloads/KalypsoNA/ and the source code of the modified part of the model presented in this paper can be provided upon request to the corresponding author."*

*This doesn't fit with the journals code availability policy. Code is recommended to sit in a* 170 *repository such as zenodo. It's also duplicated at the end of the document so could probably be removed at this point in the text."*

Answer to comment 10:

We totally agree in that. The preprint was published on 4$^{th}$ of May and the code was sucessfully published as open source code under the following link in the TORE 175 system of our university: https://doi.org/10.15480/882.3522 at the same time. But the GMD procedure didn't give us the possibility to adjust the text and add the link.

*"Line 408: "(1) applicable to model complex drainage systems in tidal backwater affected lowlands," The application is to one test case and the backwater profile is assumed flat. I don't think this is sufficient to claim applicability to all complex drainage systems – especially those with greater tidal ranges and long backwater profiles. The authors might disagree but I think this need to be a more nuanced conclusion recognising potential* 185 *limitation of the approach and the summary needs to include a critical view on the limitation of the method."*

Answer to comment 11:

Thank you for the comment. We agree in the need to work out a more nuanced discussion and conclusion. As described in the answer to comment Nr. 9, we will work out an additional chapter 7 to point out the limitations and applicability of the proposed
      conceptual method.

*Line 410: "(3) open for further model development" depends on code availability section,*
      *don't claim if not open."*

      Answer to comment 12:

      As explained in the answer to comment nr. 10: the availabilty of the code is given since
      May 2021, but it wasn't possible to change the text in the preprint.

      *"This article presents a new method for including a simple backwater effect in hydrological*
      *models that might act as a quick substitute for full hydrodynamic simulation in some lowland*
*systems. The study is well motivated and generally well presented. The methods get a bit tricky to*
      *follow in places due to being split between the supplement and main text, but the authors might*
      *have good reasons behind this.*

      *The performance of the new model is evaluated for a test case on the Dove-Elbe and shows*
      *promising results. My only significant issue is that the assumptions made by the method relative*
*to taking a hydrodynamic approach are not really discussed in detail and the conclusions thus find*
      *that the model is generally applicable to all lowland settings and scales – I think this is unlikely to*
      *be the case. "*

      Answer to comment 13:

      Thank you for the comments, remarks and description of your point of view. We agree in
your comments and are open for improving our preprint. The requirement for a more
      nuanced conclusion and a more detailed discussion about the limitations of our
      conceptual method will be added in an additional chapter. (see answer to the comments: 9
      and 11)

      As described in the answers to the comment nr. 8: Two paragraphs about the
mathematical description of the method are shifted back from the supplementary to the
      main text.

---

## Author Response (AR2)

**Response by the author: Sandra Hellmers**

**24.11.2021**

Dear Jeff,

thank you for the two comments. I am pleased to revise section 7 with the aim to improve the readability. You will see the changes in the "track-changes" file. Your second comment refers to a request from my side to add some more information about the published code. I am happy to complete that.

Best wishes,

Sandra

**Comments to the author by the handling topical editor**:
**23.11.2021**

Dear Sandra,

Thank you for your revised submission. In my opinion the scientific issues raised by the reviewers have been addressed and the article is in principal suitable for publication. I have a few minor editor revisions to overcome before publication.

1) Please have a carful read of the new text in section 7, this makes sense but the readability could be smoother.

2) As indicated by email please add links for the user manual and as with the code availability note that a 'frozen' version is needed with long term access for the instance of the code published here. You can also add links to live/developer versions if you wish.

Best wishes,
Jeff